# Optimization of Siderophore Production in Three Marine Bacterial Isolates along with Their Heavy-Metal Chelation and Seed Germination Potential Determination

**DOI:** 10.3390/microorganisms11122873

**Published:** 2023-11-27

**Authors:** Mounika Sarvepalli, Aditya Velidandi, Narasimhulu Korrapati

**Affiliations:** Department of Biotechnology, National Institute of Technology Warangal, Warangal 506004, Telangana, India; mouni.sarvepalli@gmail.com (M.S.); aditya.velidandi@gmail.com (A.V.)

**Keywords:** *Bacillus taeanensis*, marine bacteria, *Pseudomonas mendocina*, *Enterobacter*, siderophores

## Abstract

Siderophores are low-molecular-weight and high-affinity molecules produced by bacteria under iron-limited conditions. Due to the low iron (III) (Fe^+3^) levels in surface waters in the marine environment, microbes produce a variety of siderophores. In the current study, halophilic bacteria *Bacillus taeanensis* SMI_1, *Enterobacter* sp., AABM_9, and *Pseudomonas mendocina* AMPPS_5 were isolated from marine surface water of Kalinga beach, Bay of Bengal (Visakhapatnam, Andhra Pradesh, India) and were investigated for siderophore production using the Chrome Azurol S (CAS) assay. The effect of various production parameters was also studied. The optimum production of siderophores for SMI_1 was 93.57% siderophore units (SU) (after 48 h of incubation at 30 °C, pH 8, sucrose as carbon source, sodium nitrate as nitrogen source, 0.4% succinic acid), and for AABM_9, it was 87.18 %SU (after 36 h of incubation period at 30 °C, pH 8, in the presence of sucrose, ammonium sulfate, 0.4% succinic acid). The maximum production of siderophores for AMPPS_5 was 91.17 %SU (after 36 h of incubation at 35 °C, pH 8.5, glucose, ammonium sulfate, 0.4% citric acid). The bacterial isolates SMI_1, AABM_9, and AMPPS_5 showed siderophore production at low Fe^+3^ concentrations of 0.10 µM, 0.01 µM, and 0.01 µM, respectively. The SMI_1 (73.09 %SU) and AMPPS_5 (68.26 %SU) isolates showed siderophore production in the presence of Zn^+2^ (10 µM), whereas AABM_9 (50.4 %SU) exhibited siderophore production in the presence of Cu^+2^ (10 µM). Additionally, these bacterial isolates showed better heavy-metal chelation ability and rapid development in seed germination experiments. Based on these results, the isolates of marine-derived bacteria effectively produced the maximum amount of siderophores, which could be employed in a variety of industrial and environmental applications.

## 1. Introduction

Among micronutrients, iron (Fe), as a metal ion, is a much needed trace element, which is irreplaceable because of its role in several fundamental biochemical processes associated with the development and growth of all living organisms (microbes, animals, and plants) [1]. Compared with other metal ions, Fe has a unique redox chemistry and coordination [1]. By virtue of its oxidative states (ferric (Fe^+2^) and ferrous (Fe^+3^)), Fe is a part of various biochemical systems, such as transcriptional regulation, chemical transitions, providing protection against reactive oxygen species, as a cofactor in DNA synthesis, electron transport chain, bio-catalysis, biosynthetic, biodegradation, and cellular respiration pathways [1,2,3,4]. 

At acidic pH (pH < 7), Fe^+2^ is the most prevalent species in anaerobic conditions, whereas it is comparatively more soluble and accessible to living organisms in aerobic conditions; however, in aerobic conditions, it is easily oxidized into Fe^+3^, which subsequently precipitates [5]. On the other hand, at physiological pH (7.0–7.4), Fe^+3^ (10^−18^ M) is predominantly seen, but it is not readily accessible in the absence of a chelating agent [6]. Thus, Fe^+3^ is considered a primary growth limiting factor for the majority of living organisms at physiological pH [4,7,8]. 

At pH~8, in oxygenated seawater, inorganic dissolved Fe is most thermodynamically stable when it forms Fe^+3^–hydroxide complexes, which in turn reduces Fe (10^−36^ M) bioavailability [8,9]. Additionally, Fe^+3^–hydroxide complexes are present in equilibrium with Fe^+3^–oxyhydroxide complexes, which are considered to be poorly soluble. However, Fe^+3^–hydroxide complexes have the propensity to be scavenged by sinking particulate particles [9]. In comparison with their terrestrial counterparts, marine bacterial siderophores demonstrate different characteristics [10,11]. By virtue of its unique chemical composition, dissolubility kinetics, low bioavailability, and low solubility, Fe absorption/uptake is challenging [7]. 

As part of their coping mechanisms, microbes have developed several unique strategies for absorbing Fe from the surrounding habitats; among them is the production and secretion of Fe^+3^-chelating molecules, widely known as “siderophores” [6]. Siderophores are Fe-scavenging low-molecular-weight ligand (secondary metabolites) molecules (500–1500 Da), endowed with and named after their specific and high affinity for Fe^+3^ (Kf > 1030) [2,12,13]. Excreted siderophores can chelate even minute concentrations of Fe^+3^ from the habitat in order to facilitate uptake by microbes, thus preventing their precipitation and loss of its bioavailable forms [14]. In general, Fe^+3^ can be reversibly bound by siderophores; thus, their transport, uptake, reactivity, and bioavailability can be closely regulated [14,15]. 

Under Fe^+3^-deficient growth conditions, the hydrophobicity of the microbial surface is greatly decreased, leading to changes in the composition of surface proteins followed by limitation of biofilm formation, which then ultimately results in microbial death [16]. Therefore, during the low bioavailability of Fe^+3^ in the surrounding habitat, siderophores are produced and secreted by the microbes; to date, it is one of the evolutionary coping mechanisms developed by microbes to survive in a Fe-deficient habitat [16]. With the help of siderophores, microbes can uptake Fe in the form of Fe^+3^–siderophore complexes, and then, in the cytosol, Fe is reduced and released as Fe^+2^ [3]. Fe^+3^ forms a hexadentate octahedral complex with the siderophores during their transportation into the microbes [2]. Upon the disassociation of Fe^+2^–siderophore complexes inside the microbes, siderophores are released and recycled again for Fe^+3^ acquisition [5]. 

Since siderophores can greatly increase the amount of dissolved Fe^+3^ by solubilizing the Fe particulates (insoluble), this in turn will affect the bioavailability of Fe in the environment, thus shaping the composition and structure of the microbial ecosystem [17]. Additionally, this will greatly affect the bioavailability of Fe^+3^ to non-siderophore-producing microbes, thus acting as a key regulator in the bioavailability of Fe and other biogeochemical cycles [17]. 

Depending on membrane affinity and length, siderophores could anchor a particular gradient outside the microbial membrane, thereby increasing the efficiency of siderophores in chelating Fe^+3^ from the environment [16]. On the other hand, Fe^+3^ acquisition is closely regulated by the microbes, as the occurrence of intracellular Fe^+3^ in excess amounts will result in oxidative stress, leading to microbial death. In this regard, siderophores also act as key regulators in microbial Fe homeostasis [8]. In comparison with other metal ions, Fe^+3^ is preferentially chelated by siderophores [16,18]. Siderophores can be further classified into carboxylates, catecholates, hydroxamates, and mixed-types siderophores depending on the primary oxygen-donating ligands, which bind with Fe^+3^ [2]. 

Numerous siderophores have been isolated from marine bacteria, including *Actinomycetes*. The *Streptomycetes* group of organisms are well known for their production of numerous siderophores, which act contingently and are regulated independently to compete efficiently in the environment. *Streptomycetes* are a highly available and important group of actinomycetes in the marine environment [19]. Actinobacteria from both aquatic and soil sources have been widely reported in the production, transportation, and regulation of deferoxamines (hydroxamate siderophore) [20]. Some siderophores isolated from actinobacteria reduce the cadmium uptake by bacteria. Under nickel contamination, *S. acidiscabies* E13 secreted the hydroxamate type of siderophores. This promoted cowpea growth by binding nickel and iron, inhibited the uptake of nickel, and supplied iron for plant growth [21].

Heavy metals severely hamper the ecosystems of the terrestrial and aquatic realms. Finding long-lasting ways of eliminating these harmful substances from the environment is necessary due to their persistence in food chains and ecological niches. The persistence of heavy metals reduces the time they take to enter the food chain through sources such as aquatic plants, fish, and other aquatic creatures. Heavy-metal accumulation in the aquatic environment, particularly during the early phases of the development of fish, can be lethal to many different kinds of fish [22]. Heavy-metal contamination of soil has significant effects on the ecosystem, making it a critical environmental problem. The heavy metals in the soil can enter human food via plants, posing risks, since they are passed down the food chain [23]. With heavy metals posing considerable risks to humans and the environment, it is imperative that heavy-metal-polluted soils be decontaminated immediately [24]. It is possible that siderophore chelators may play a critical role in bacteria resistance to heavy metals if toxic metals induce the production of siderophores [25]. 

Bacteria, which produce siderophores, can promote the growth of plants, also protecting them against fungal infections, as well as degrading pesticides. Thus, siderophores can benefit plants and help combat iron deficiency in soil microbes in tough environments. They can enhance the bioavailability of iron and benefit the roots of plants [3]. However, the exact mechanism is not well understood. A couple of possible mechanisms have been proposed for plants’ acquisition of iron from microbial siderophores. First, plant transport systems can receive Fe^+2^ from the siderophores of micro-organisms with a high redox potential, and second, microbial siderophores have the ability to chelate Fe from soils and subsequently exchange ligands with phytosiderophores [26]. 

Based on the above-stated information, the objectives of the present work were (i) to identify the unexplored sample source (Kalinga beach (17°85′53″ N, 83°41′71″ E), Bay of Bengal, Visakhapatnam, India), (ii) to isolate and screen the siderophore-producing marine bacteria using the Chrome Azurol Sulfate (CAS) assay, (iii) to perform 16sRNA molecular characterization of marine bacteria, (iv) to study the effect of various process parameters (incubation time, temperature, initial pH, carbon source, nitrogen source, organic acids, and metal ions) of siderophore production, (v) to use marine bacterial isolates as chelators for various heavy metals, and (vi) to use a cell-free supernatant for seed germination.

## 2. Materials and Methods

### 2.1. Sample Collection 

The marine water and soil samples were collected from Kalinga beach (17°85′53″ N, 83°41′71″ E), Bay of Bengal, Visakhapatnam (Andhra Pradesh, India). The soil samples were collected at 2 m depth from the surface, and the water samples were collected 10 m away from the coast. The samples were then transferred into sterile poly propylene screw cap bottles, polythene bags, and they were preserved at 4 °C for future studies.

### 2.2. Isolation and Screening of Siderophore-Producing Marine Bacteria

To isolate the siderophore-producing marine bacteria, 1 g of the soil sample was dissolved in 10 mL of distilled water (DH_2_O) and kept under shaking for 1 h. Water and soil samples were serially diluted, and 10^−4^–10^−6^ dilutions were spread on Zobell Marine Agar 2216 (ZMA) plates and incubated at 30 °C for 72 h [27]. All the isolated colonies were sub-cultured repeatedly on ZMA to obtain pure cultures. The obtained pure cultures were stored at 4 °C, and 30% glycerol stocks were prepared and preserved at −20 °C for future use [28].

The isolated bacteria were tested for siderophore production, as per the following procedure. Chrome Azurol S (CAS) agar plates were prepared with 2 mM of CAS in 50 mL DH_2_O, 10 mL of 1 mM ferric chloride in 10 mM hydrochloric acid (HCl) and slowly added to 40 mL of 72.9 mg of hexadecyl trimethylammonium bromide (HDTMA), which resulted in deep blue color solution. The final solution was then autoclaved. The mixture was prepared with 30.24 g of piperazine in 800 mL of DH_2_O, 100 mL of production medium, 15 g/L of Agar, pH 5.6 (adjusted using conc. HCl) and autoclaved separately. This mixture was then added to the above dye solution under continuous stirring to avoid foam formation and aseptically transferred to sterile Petri plates. All bacterial isolates were streaked on CAS agar plates and incubated for 72 h at 28 °C [21].

### 2.3. Characterization of Siderophore-Producing Marine Bacteria

The bacterial isolates, which showed the maximum orange-yellow halo zone, were identified based on characteristics according to *Bergey’s Manual of Systematic Bacteriology* and confirmed with 16S ribosomal ribonucleic acid (rRNA) gene identification. The 16S rRNA gene (1500 bp) [29] was amplified with Polymerase chain reaction, and the Exonuclease I-Shrimp Alkaline Phosphatase was used to purify the amplicons, which were then sequenced using instrument ABI 3500xL genetic analyzer (Life Technologies, Carlsbad, CA, USA) following Sanger’s method [30]. The closest culture sequence was collected from the National Centre for Biotechnology Information database, and the bacterial sequences were analyzed using the Basic Local Alignment Search Tool (BLASTn), which detects region similarity between sequences [31]. MEGA 10.2.2 version (www. megasoftware.net, Accessed on 25 October 2023) was used to construct phylogenetic trees by using the neighbor-joining algorithm.

### 2.4. Growth Curve Estimation

The siderophore-producing bacterial cultures were inoculated in 100 mL of marine broth and incubated at 30 °C for 72 h in a shaking incubator. To measure the growth of isolates, optical density (OD_600_) was measured in quartz cuvettes (Hellma analytics, Müllheim, Germany) using a UV-Vis spectrophotometer (Shimadzu UV-1800 Spectrophotometer, Kyoto, Japan).

### 2.5. Siderophore Production and Estimation

The culture was inoculated into Fe-free succinate medium (SM) (4 g/L succinic acid, 6 g/L K_2_HPO_4_, 3 g/L KH_2_PO_4_, 0.2 g/L MgSO_4_.7H_2_O, 1 g/L (NH_4_)_2_SO_4_, initial pH 7.5) and incubated for 48 h at 28 °C and 180 rpm [32]. To remove traces of iron, all the glassware was washed with 6 M HCl prior to use. The culture was later centrifuged at 10,000 rpm for 10 min at 4 °C, and the supernatant was stored at 4 °C until further use. Siderophore production was estimated with the CAS assay. To 0.5 mL of the CAS assay solution, 0.5 mL of the supernatant was added, and absorbance was measured at 630 nm. The production of siderophores was calculated with the following standard formula (Equation (1)) [33] and expressed as siderophore units (%SU).
(1)$$\%\mathrm{SU}\;(\mathrm{siderophore}\;\mathrm{units})=\left(\frac{A_{r}-A_{s}}{A_{r}}\right)\times 100$$
where “*A_r_*” is the reference absorbance, and “*A_s_*” is the sample absorbance at 630 nm.

To determine the chemical nature of siderophores, tetrazolium and Arnow assays were performed [34,35]. The tetrazolium test was used to detect the hydroxamate type of siderophores [34]. To 100 µL of the supernatant, a pinch of tetrazolium salt and 1–2 drops of 2 N NaOH were added. The immediate emergence of an intense red hue indicates the presence of a hydroxamate-type siderophore. The occurrence of a catecholate-type siderophore is identified with the Arnow test. To 1 mL of the culture supernatant, 0.1 mL of 0.5 M HCl followed by 1 mL of nitrate molybdate reagent and 1 mL of NaOH were added [35]. The presence of catecholate in the supernatant is indicated by a color shift from yellow to vivid orange red after 5 min of incubation.

### 2.6. Effect of Production Parameters on Siderophore Production

The production of siderophores was affected by different medium components and physicochemical parameters (such as incubation time, temperature, initial pH, carbon source, nitrogen source, amino acids, and metals). All the said parameters were studied using the one-factor-at-a-time (OFAT) approach to evaluate their effect on siderophore production and bacterial growth. The impact of incubation time on siderophore production was studied at intervals of 12 h (at 12, 24, 36, 48, 60, and 72 h). The effect of temperature on siderophore production was studied at 20, 25, 30, 35, 40, and 45 °C. The impact of pH of the medium was studied by adjusting the pH to 6.5, 7, 7.5, 8, 8.5, and 9 using 1 N NaOH. 

The effects of various carbon (0.1% *w*/*v*) sources (such as sucrose, glucose, maltose, fructose, and xylose) and nitrogen (0.1% *w*/*v*) sources (such as sodium nitrate, ammonium sulfate, yeast extract, peptone, and urea) were studied. The influence of organic acids (0.2% *w*/*v*) on siderophore production was studied by using succinic acid, citric acid, and oxalic acid. The effect of different concentrations of Fe^+3^ (0.01, 0.10, 1.00, and 10.0 µM) was also studied. Lastly, the effect of different metal ions (Fe^+3^, Cu^+2^, Mn^+2^, and Zn^+2^) on the production of siderophores was also determined by supplementing the production medium with FeCl_3_, CuSO_4_, MnSO_4_, and ZnSO_4_ salts at a concentration of 10 µM. Siderophore production during the experiments was determined using the CAS assay, as discussed earlier, and cell growth was measured at OD_600_ using a UV-Vis spectrophotometer.

### 2.7. Heavy-Metal Chelation

The ability of siderophore-producing marine bacteria to chelate different metal ions was tested using the CAS agar plate method by spotting the bacterial cultures on it. As discussed earlier, the CAS + Fe^+3^ (Metal) + HDTMA plates were prepared by replacing Fe^+3^ with heavy-metal ions, such as Ag^+2^, Al^+2^, Cd^+2^, Co^+^, Cr^+6^, Hg^+2^, La^+3^, Mo^+6^, Ni^+2^, Pb^+2^, Pd^+2^, and Y^+3^. The concentration of all metal ions was kept to 1 mM metal stock in 10 mM conc. HCl. Each modified CAS plate was inoculated with a logarithmic phase culture of marine bacterial isolates and incubated for 72 h at 28 °C [36,37].

### 2.8. Seed Germination

To evaluate the effect of marine bacterial siderophores on seed germination, Brown chickpea (*Cicer arietinum* L.), Peanut (*Arachis hypogaea*), Green gram (*Vigna radiata*), and Kabuli chana (*Cicer arietinum*) were collected from a local market, and healthy seeds of similar sizes and shapes were selected for further experiments. The seeds were surface-sterilized by being soaked in 75% of ethanol, repeatedly washed with sterile deionized water to eliminate any chemicals, and dried using tissue paper. Fifty seeds of each type were selected and soaked in the cell-free supernatant of the siderophore production medium in a Petri dish and incubated at 25 °C for 36 h. The seedlings were monitored and considered germinated when the root protruded from the seed coat by at least 2 mm. Five seedlings per plate were randomly selected to determine the root lengths. The germination percentage was determined using Equation (2) [38,39].
Germination percentage (% GP) = (Seeds germinated/Total seeds) × 100 (2)

### 2.9. Statistical Analyses

All experiments were performed in triplicate, and the error bars reflect the standard deviation ± mean. Statistical analyses were carried out using Microsoft Excel 2019. A *p*-value < 0.05 was considered statistically significant.

## 3. Results

### 3.1. Identification and Characterization of Siderophore-Producing Marine Bacterial Isolates 

ZMA medium was used to isolate sixty-eight marine bacteria from Kalinga beach, Bay of Bengal (Visakhapatnam, Andhra Pradesh, India). The pH and temperature of marine samples were 8.19 and 24 °C. Among them, 70% of the isolates had siderophore-producing ability, which was confirmed by observing the “orange halo zones” on blue agar plates. 

Three marine bacterial isolates (SMI_1, AABM_9, and AMPPS_5) showed higher siderophore production compared with the other marine bacterial isolates. All three marine bacterial isolates (SMI_1, AABM_9, and AMPPS_5) were identified with colony morphology, biochemical tests, and molecular characterization (Table 1). SMI_1 showed white circular and creamy colonies, which were Gram-positive in nature. SMI_1 showed positive for oxidase and catalase, whereas it rendered a negative result for hydrolysis of casein and gelatin. The AABM_9 colonies were circular, smooth edged, convex, and dull yellow colored. AABM_9 exhibited Gram-negative, rod-shaped, halotolerant bacteria and showed positive for catalase and hydrolysis of gelatin, but it showed negative for oxidase and hydrolysis of casein. AMPPS_5 showed flat, non-wrinkled, and pale brownish yellow colonies. AMPPS_5 exhibited rod-shaped, Gram-negative bacteria. AMPPS_5 showed positive for catalase, oxidase, and gelatin hydrolysis, whereas it rendered a negative result for casein hydrolysis. 

The 16S rRNA sequencing was performed for molecular identification of the isolates. The BLAST results showed that SMI_1 was closely related to *Bacilllus taeanensis*, showing 98% similarity (Figure 1). The 16S rRNA gene sequence of AABM_9 showed closest similarity with *Enterobacter hormaechei*, with 97.47% in the NCBI BLAST analysis (Figure 2). AMPPS_5 was closely related to *Pseudomonas mendocina*, with 99% similarity in the NCBI database (Figure 3). The 16S rRNA sequences of SMI_1, AABM_9, and AMPPS_5 marine bacterial isolates were deposited at GENBANK with accession numbers MW375467, MW535739, and MW444999, respectively. The phylogenetic trees for evolutionary analysis were generated with the neighbor-joining method. The values adjacent to the branches reflect the percentage of replicate trees in which similar taxa were grouped in a 1000-replica bootstrap test.

### 3.2. Production and Estimation of Siderophores

The siderophore production medium was used to produce siderophores. On CAS agar plates, the orange halo zones were formed around the isolates; the CAS–Fe^+3^–HDTMA complex was fragmented by siderophores and formed the Fe^+3^–siderophore complex, and CAS dye was released, which resulted in orange zone formation [40,41]. The amount of siderophore produced was determined by inoculation of the isolates in the production medium, followed by incubation at 28 °C for 48 h; the change in the medium color from colorless to golden yellow indicates the occurrence of siderophores in the medium, known as “siderophoregenesis” [40]. The CAS assay was performed to quantify siderophore production using Equation (1). SMI_1 produced 59.8 (%SU), AABM_9 produced 60.38 (%SU), and AMPPS_5 produced 50.6 (%SU). The SMI_1 and AABM_9 isolates showed a positive result in the Arnow test, which reflected the presence of the catecholate type of siderophore, whereas AMPPS_5 showed positive in both Arnow and tetrazolium tests, indicating the occurrence of a mixed-type siderophore (both catecholate and hydroxamate).

### 3.3. Effect of Process Parameters on Siderophore Production

In the literature, several factors, such as incubation time, temperature, initial pH, carbon source, nitrogen source, organic acids, and metal ions, have been reported to influence siderophore synthesis and secretion [6]. Furthermore, the conditions may vary greatly depending on the type of bacterial species and strain used to achieve maximum siderophore production [42]. Therefore, based on the above-stated information, to enhance siderophore production from marine bacterial isolates (SMI_1, AABM_9, and AMPPS_5), different siderophore production process parameters and their influence were investigated using the one-factor-at-a-time (OFAT) approach. Table 2, Table 3 and Table 4 present the effect of various physicochemical parameters on the siderophore production (%SU) of marine bacterial isolates SMI_1, AABM_9, and AAMPS_5, respectively. Table 5 presents a comparative analysis of siderophore production of the marine bacterial isolates employed in this study and those reported in the literature.

#### 3.3.1. Incubation Time

Siderophores were mostly produced and secreted during the stationary phase. SMI_1 showed maximum production (60.17 %SU) at 48 h of incubation time, whereas AABM_9 (65.68 %SU) and AMPPS_5 showed maximum production (58.86 %SU) at 36 h. AMPPS_5 started producing siderophores from the late log phase after 12 h. Bacteria with a long log phase and maximum siderophore production showed optimum siderophore production during the late log phase and early stationary phase. This reflects the essential requirement for Fe^+3^ for bacterial growth. The trend of siderophore production followed similar bacterial growth curve trends of the isolates, as shown in Figure 4a. 

#### 3.3.2. Temperature

The influence of temperature on siderophore production and growth of isolates was also investigated. It was observed that marine bacterial isolates were stable in a range of temperatures. The isolates were investigated in temperatures from 20 °C to 45 °C, with the highest amount of yield observed at 30 °C (65.45 %SU), 30 °C (68.89 %SU), and 35 °C (64.05 %SU) for SMI_1, AABM_9, and AMPPS_5, respectively (Figure 4c). 

#### 3.3.3. Initial pH

Consequently, pH is one of the essential parameters, which can affect the production of siderophores. Variations in ambient pH can also affect Fe^+3^ bioavailability and growth of the organism. The optimum initial pH for a maximum yield of siderophores for SMI_1 and AABM_9 was pH 8 (74.91 %SU and 82.5 %SU), whereas for AMPPS_5, it was pH 8.5 (69.65 %SU) (Figure 4e).

#### 3.3.4. Carbon Source 

The effect of different carbon sources on siderophore production was investigated. Carbon (0.1% *w*/*v*) sources such as glucose, fructose, sucrose, maltose, and xylose were used, and the optimized physical parameters were maintained for every individual isolate for further production. SMI_1 and AABM_9 produced maximum siderophores in the presence of sucrose (78.91 %SU and 76.4 %SU), whereas AMPPS_5 showed maximum production with glucose (75.69 %SU). As carbon sources, both glucose and sucrose strongly influenced the growth of marine bacterial isolates (Figure 5a). 

#### 3.3.5. Nitrogen Source

The influence of various nitrogen (0.1% *w*/*v*) sources (inorganic and organic forms) on the generation of siderophores was investigated. Ammonium sulfate ((NH_4_)_2_SO_4_), sodium nitrate (NaNO_3_), yeast extract, peptone, and urea were used as nitrogen sources. SMI_1 (84.53 %SU) showed higher production in NaNO_3_, whereas AABM_9 (82.54 %SU) and AMPPS_5 (82.404 %SU) showed higher production in (NH_4_)_2_SO_4_ (Figure 5c). 

#### 3.3.6. Organic Acids

The effect of different organic acids on siderophore production was evaluated. Organic acids (0.2%) such as succinic acid, oxalic acid, and citric acid were used. SMI_1 (91.74 %SU) and AABM_9 (64.49 %SU) produced the highest amount of siderophores in the presence of succinic acid, whereas AMPPS_5 (87.25 %SU) showed optimum production in the presence of citric acid (Figure 5e). Later, the influence of different concentrations of succinic acid and citric acid on SMI_1 (Figure 5g) and AMPPS_5 (Figure 5g) isolates was investigated, respectively. SMI_1 (93.57 %SU) and AABM_9 (85.9 %SU) showed maximum siderophore production with 0.4% succinic acid, whereas AMPPS_5 showed maximum siderophore production with 0.4% citric acid (91.17 %SU). It was also observed that as the concentrations of organic acids increased from 0.4% to 1.0%, there was a steady decrease in siderophore production. This might be due to the presence of excessive organics acids in the medium, resulting in the inhibition of siderophore production. 

#### 3.3.7. Concentration of Iron (Fe^+3^)

The effect of different concentrations (0.01, 0.10, 1.00, and 10.0 µM) of Fe^+3^ on the marine bacterial isolates was evaluated. Based on the results, SMI_1 showed maximum siderophore production at 0.10 µM concentration, whereas AABM_9 and AMPPS_5 showed maximum siderophore production at 0.01 µM concentration (Figure 6a). 

#### 3.3.8. Different Metal Ions

Based on the above-stated explanations, the effect of different metal ions, such as Fe^+3^, Cu^+2^, Mn^+2^, and Zn^+2^, at 10 µM on siderophore production was evaluated. SMI_1 showed maximum siderophore production with Zn^+2^ (73.09 %SU) followed by Mn^+2^ (71.70 %SU), Cu^+2^ (58.21 %SU), and Fe^+3^ (43.68 %SU) (Figure 6c). AABM_9 showed maximum siderophore production with Cu^+2^ (50.41 %SU) followed by Zn^+2^ (47.78 %SU), Mn^+2^ (45.20 %SU), and Fe^+3^ (36.24 %SU). Meanwhile, AMPPS_5 showed maximum siderophore production with Zn^+2^ (68.26 %SU) followed by Cu^+2^ (62.01 %SU), Mn^+2^ (52.97 %SU), and Fe^+3^ (38.57 %SU) (Figure 6c). 

### 3.4. Heavy-Metal Chelation

After 72 h of incubation, a color change from blue to yellow was observed, which reflected the chelation of the metal, and a CAS plate with Fe^+3^ was considered as control. SMI_1 yielded strongly positive results in Co^+2^, Cr^+6^, Hg^+2^, and Ni^+2^, whereas positive results were obtained in Ag^+2^, Al^+2^, La^+3^, Mo^+6^, Pd^+2^, and Y^+3^. SMI_1 did not show any activity in Cd^+2^ and Pb^+2^, which reflected the absence of siderophore production. AABM_9 showed clear zone formation in Al^+2^, La^+3^, Pd^+2^, and Y^+3^ and showed moderate zone formation in Cd^+2^, Co^+2^, Cr^+6^, Hg^+2^, Ni^+2^, and Pb^+2^. No activity was observed in Ag^+2^ and Mo^+6^. AMPPS_5 showed moderate chelation in Al^+2^, Cd^+3^, Co^+2^, Cd^+2^, Hg^+2^, La^+3^, Ni^+2^, and Pb^+2^. It showed weak chelation in Pd^+2^ and Y^+3^, whereas a negative result was observed in Ag^+2^ and Mo^+6^ (Figure 7 and Table 6).

### 3.5. Seed Germination

In this study, the effect of siderophores on the germination of seeds (Brown chickpea (*Cicer arietinum* L.), Peanut (*Arachis hypogaea*), Green gram (*Vigna radiata*), and Kabuli chana (*Cicer arietinum*)) was investigated. Seeds were incubated in the cell-free supernatant of the siderophore production medium, and seeds incubated in water were used as control. Before experimentation, the supernatant was examined for the presence of siderophores using the CAS assay. The siderophore-positive supernatant was used for further work. The germination percentage was calculated after 36 h, but sprouting was observed regularly, and the length of the root was measured every 12 h. After 36 h, the growth of seedlings and the increase in plumule length were observed in all bacterial isolates (Figure 8). Among the three isolates, AABM_9 showed better seedling growth than SMI_1 and AAMPS_5, and a high germination percentage was observed in SMI_1 followed by AABM_9 and AAMPs_5 compared to control. The root length and germination percentage of each seed with the respective isolate supernatant are reported in Table 7.

## 4. Discussion

Unsurprisingly, most of the bacteria produce siderophores under Fe-deficient conditions. Marine-derived bacteria are eminent in the production of significant bioactive molecules and peptides, which have industrial and pharmaceutical importance [4,6,53]. Compared with terrestrial siderophores, most of the marine siderophores show unique structural properties [11]. In view of that, they show photochemical and amphipathic properties in Fe^+3^ complexes [16]. Marine bacterial siderophores were reported to have suites of amphiphilic siderophores with a series of fatty acid attachments to help with limited diffusion and the retainment of siderophores. When Fe^+3^ coordinated with the α-hydroxycarboxylic acid head group, which includes citrate or β-hydroxyaspartate, the siderophores reflected the photoreactive nature. Another emerging category of marine siderophores is the triscatechol siderophore, which contains catechol 2,3-dihydroxybenzoic acid (DHB). Marine bacteria, which produce triscatechol siderophores, such as enterobactin (several other triscatechol siderophores have been reported), enable Fe^+3^ to form complexes with three 2,3-DHB [54].

Liu et al. proposed that *B. taeanensis* should be reclassified into a new genus named *Maribacillus* [55]. However, it was not updated in NCBI; therefore, the authors considered *B. taeanensis* as the isolate in the present work. Halophilic *B. taeanensis* was first isolated from soil in a solar saltern in Korea [27]. Palleroni et al. first reported the identification of *P. mendocina* bacterial isolate from water and soil samples for siderophore production [56]. Later, *P. mendocina* was reported to have a potential role in bioremediation to degrade toluene [57] and oil degradation [58]. Halophilic *B. taeanensis* was first isolated from soil in a solar saltern in Korea [27].

Three *Bacillus* sp., namely *B. anthracis, B. thuringiensis,* and *B. cereus,* were reported to produce the catecholate type of siderophores, such as petrobactin and bacillibactin [59]. Wu et al. reported that marine sponge-associated *Bacillus* sp. produces two rare marine siderophore bacillibactins E and F, which contain nicotinic acid and benzoic acid moieties [60]. The Enterobacteriaceae family produces the characteristic catecholate type of siderophores, such as enterobactin, aerobactin, and yersinabactin [61]. Pyoverdine was excreted by the rhizobacteria *P. putida* KNUK9 in a Fe^+3^-deficient case and had an antagonistic effect against *Aspergillus niger* [62]. The pathogenic *P. aeruginosa* was reported to produce two different siderophores, pyochelin and pyoverdine [63]. Very few studies were reported on *P. mendocina*, whereas no reports were found on *B. taeanensis*.

The time required for the maximum production of siderophores varies greatly from strain to strain and species to species [64]. In this regard, it is of great significance to evaluate the influence of incubation time on siderophore production. Both marine bacterial isolates were inoculated in Fe^+3^-deficient medium and incubated for 72 h, and siderophore production was estimated every 12 h. Upon further increase in the incubation period, siderophore production was reduced, perhaps due to a lack of nutrients in the medium. The maximum siderophore production from *Bacillus* sp. PZ-1 was reported at 48 h [46], and in the case of *P. aeruginosa* FP6, maximum siderophore production was reported after 36 h of incubation [43], which supports the present study.

In general, temperature significantly influences siderophore production, as siderophore biosynthesis is highly temperature-sensitive [53]. The isolated bacteria in this investigation were primarily incubated at 30 °C; the optimum siderophore production was observed at 30 °C for SMI_1 and AABM_9, and 35 °C for AMPPS_5. As per the literature, the gene expression associated with siderophore production and secretion will be regulated by the temperature in which the bacteria grow [65]. It was reported that when bacteria were incubated at a temperature (optimal) closer to that observed in nature or at a sub-optimal temperature, they exhibited maximum siderophore production [4,65]. On the other hand, incubation of bacteria at a sub-lethal temperature can only result in the total loss or decrease in siderophore production [4]. It was also reported that bacteria with the ability to produce more than one siderophore in nature can only produce each type in response to different laboratory temperature conditions [65]. Upon an increase in temperature, a decrease in the production was observed. Kumar et al. reported that *B. thuringiensis* (VITVK5 strain) and *Enterobacter soli* (VITK6) isolates showed high yield at 35 °C [48]. *B. licheniformis* and *B. subtilis* showed maximum siderophore production at 25 °C, and upon increasing the temperature, they observed a sheer decline in production [7]. The temperature range varies considerably depending on the type of bacteria; for example, 25 °C was observed for *Aspergillus niger* and *Penicilium oxalicum* [66], whereas 30 °C was observed for *Anabaena oryzae* [67] and *P. fluorescens* [64], and 55 °C for *Escherichia coli*, *Bacillus* spp. ST13, and *Streptomyces pilosus* [49]. However, a study conducted by Sinha et al. on siderophores from marine bacteria reported that different temperatures affected bacteria growth but not siderophore production [44]. 

In general, maximum siderophore production is observed at neutral pH (~7) [4]. Fe solubility is greatly dependent on pH [53]. It was observed that several bacteria produce siderophores in the pH range of 7.0–8.0 because Fe is not physiologically soluble at this pH [68]. On the contrary, a decrease or total loss of siderophore production will be observed in highly alkaline and acidic pH conditions [4]. Alkaline pH solubilizes Fe, whereas acidic pH (<5) conditions negatively affect bacterial growth in most cases [68]. Therefore, the degree (pH) of alkalinity or acidity of the medium is a crucial factor, which can influence siderophore production as well as bacterial growth [6]. The bacterial isolates in this study were maintained at pH 7, but the maximum siderophore production was identified in the pH range of 8–8.5 in this study. The growth of marine bacterial isolates was high at alkaline pH, as was siderophore production. The maximum siderophore production from *P. fluorescens* S-11 was reported at pH 7 [47]. The maximum siderophore production was observed at pH 8–8.2, while decreased production was reported at lower pH conditions in marine bacteria [44]. Kumar et al. reported that at moderate pH (~8) conditions, *B. thuringiensis* (VITVK5 strain) and *Enterobacter soli* (VITVK6 strain) showed maximum siderophore production [48]. A study conducted by Sinha et al. showed the highest production of siderophores at pH 8.5 from different marine bacteria [44], and the present study confirms that alkaline pH is conducive to higher siderophore production.

In general, various carbon sources help in the regulation of siderophore production in bacteria [53]. In bacteria, the specific amino moieties of different siderophores are the derivatives of succinate, malate, or α-ketoglutarate [43,69]. Therefore, the media with different types of carbon sources show varying rates of siderophore production [48,70]. Additionally, the carbon source will also determine the quality of the media and bacterial metabolism, since they are essential for bacterial growth and siderophore production [6]. Within the medium composition, the selection of a suitable carbon source is of great significance [64]. Both glucose and sucrose are simple carbohydrates and can be quickly absorbed by bacterial isolates, resulting in increased production of siderophores. Xylose was significantly less suitable for producing siderophores and growth. It was also reported that glycerol (as a carbon source) enhanced siderophore production in *B. megaterium* [42], *Bacillus* spp. [46], and *E. coli* [71] compared to maltose, lactose, and galactose. *P. aeruginosa* FP6 showed increased siderophore production in the presence of sucrose and mannitol as a carbon source [43]. Kalyan et al. reported that *B. licheniformis*, *B. subtilis*, and *Ochrobactrum grignonense* isolates showed the highest siderophore production in sucrose [7].

Nitrogen source is another important parameter, which will greatly influence siderophore production along with bacterial growth [53]. Similar to the carbon source, nitrogen source will also affect the medium quality and bacterial metabolism [6]. On the other hand, the evaluation of the influence of nitrogen source on siderophore production has shown contradictory results, which were dependent on the type of nitrogen form used (inorganic: sodium nitrate, ammonium nitrate; organic: urea, peptone, and yeast extract) [4]. It was also evident that both marine bacterial isolates prefer inorganic forms of nitrogen source over organic forms. Urea also stimulated siderophore production in both isolates. Similar results were reported in siderophores produced from *B. substilis* DR2, which showed maximum production when using NaNO_3_ as a nitrogen source [28]. The presence of yeast extract and urea stimulated the production of siderophores in *P. aeruginosa* [43]. Few studies have reported the use of amino acids as nitrogen sources. For example, supplementing the medium with glutamic acid or asparagine has shown increased siderophore production in *Microbacterium* sp. [72].

In addition to the carbon and nitrogen sources, the choice of organic acids used will also affect siderophore production [53]. For example, siderophores such as aerobactin, rhizobactin, and schizokinen require organic acids as their derivatives [73]. *P. aeruginosa* RZS9 showed maximum production in the presence of 5 g/L succinic acid in statistical and L-capacity reactor experiments [50]. Few studies have also reported the influence of succinic acid and citric acid on siderophore production from *Achromobacter* sp. [73]. and *Pseudomonas* spp. [48], respectively.

As stated in the Introduction, the concentration of Fe^+3^ in the culture medium plays a vital role in the promotion of transcriptional factors essential for the biosynthesis of siderophores [4]. In general, siderophore production only occurs if the bacteria are under Fe^+3^-deficient conditions [53]. In addition, even at low concentrations, Fe^+3^ is essential for bacterial growth and survival [4]. However, the minimum concentration of Fe^+3^ required in the medium for the biosynthesis of siderophores is different for different bacteria. Furthermore, the concentration of Fe^+3^ required to inhibit siderophore production is (i) dependent on the bacteria; therefore, it varies from strain to strain and species to species and (ii) also depends on the culture medium composition [4]. In general, a Fe concentration of ≤0.1 µmol/L does not suppress siderophore production [4]. The occurrence of Fe^+3^ in concentrations beyond the threshold limit will lead to negative transcriptional regulation of the associated genes, resulting in decreased production and transport of siderophores [53].

It was evident that AABM_9 and AMPPS_5 required much lower concentrations of Fe^+3^ than SMI_1 for siderophore production. As the concentration increased to 10 µM, a significant reduction in siderophore production was observed. This indicates that higher concentrations of Fe^+3^ will inhibit siderophore production in both isolates. The results observed were, in fact, aligned with the literature [66,74,75]. On the other hand, the concentration of Fe^+3^ not only affects siderophore production but also assists in switching between multiple siderophores [53].

Recent literature suggests that siderophores are also capable of forming complexes with other metal ions beyond Fe^+3^, such as Cd^+2^, Cu^+2^, Co^+2^, Ga^+3^, In^+3^, Pu^+4^, U^+4^, Th^+4^, Al^+3^, Mg^+3^, Mn^+2^, and Zn^+2^ [53]. This indicates that siderophores are also produced and secreted even in the presence of other metal ions, not only Fe^+3^. In addition to Fe^+3^, other metal ions, such as Mg^+2^, Cu^+2^, Mn^+2^, Mo^+4^, Ni^+2^, and Zn^+2^, are essential in trace amounts for the regular functions (for example, they act as co-factors in several enzymes) of the microbes, known as “micronutrients” [43,75,76]. Based on the results, it was evident that at higher concentrations of metal ions (such as Mn^+2^, Zn^+2^, and Cu^+2^), enhanced siderophore production could be observed, which was much higher than in the presence of Fe^+3^ metal ions. The results observed were in alignment with previous reports [75,77].

Previous studies on siderophore complexation abilities with metals other than iron were reported. Trace metals, such as Cd, Cu, Ni, Pb, and Zn, in seawater form strong metal–organic complexes. Three of the five metals form a stable and firm complex with siderophore-like organic molecules, where the ligands, particularly for Cd and Ni, are substantially more potent and may have reasonably distinct functional groups [78]. Desferrioxamine B, a bacterial siderophore, increased the phytoextraction of rare earth metals, including La, Nd, Gd, Er, and Ge, by forming soluble complexes, which improved the movement of elements in the rhizosphere [79]. Arsenic-tolerant *Actinobacteria* were isolated, and a siderophore, which binds arsenic, was reported [80]. *Marinobacter* sp. SVU_3, which were isolated from a marine source, showed strong chelation of Al, Co, La, and Pb, according to one report [52].

Under a Fe limitation condition, siderophore-producing micro-organisms can be used as bio-fertilizers to increase wheat growth. Fe shortage inhibits growth and weakens photosynthesis. *Bacillus* sp. WR12 increased Fe acquisition and alleviated these effects [1]. In this study, the siderophores produced by *Enterobacter* AABM_9 significantly improved seedling length in peanuts compared to Brown chickpea, Green gram, and Kabuli chana. The growth and germination were likely significantly affected by the inoculum, but the extent of the effect varied greatly depending on the symbiosis between the seed and the given inoculant. Siderophore-producing *P. fluorescens* RB5 showed an inhibitory effect on phytopathogen *R. cerealis*, which causes wheat sheath blight [81]. Hence, the bacteria producing siderophores can be employed as a bio-fertilizer to enhance plant growth and as a bio-control agent against plant pathogens. Roslan reported the plant-growth-promoting potential of *Enterobacter* sp.; IAA-producing and P-solubilizing *E. ludwigii* improved the growth of flax culture and root surface area and improved the uptake of essential metals and Ca uptake in pigeon pea and chicken pea. *Enterobacter* sp. increased the P and K uptakes, which were reflected in the improvement in okra seedling plant [82].

## 5. Conclusions

Fe is one of the essential and abundant metals on the surface of the earth, but at physiological pH, it is unavailable to all living organisms. Similarly, in ocean water, it is accessible to microbes in minimal concentrations. To overcome Fe scarcity, the marine bacteria produce siderophores. The present study reports on the isolation, identification, and characterization of siderophore-producing marine bacterial isolates SMI_1, AABM_9, and AMPPS_5. On performing biochemical tests and molecular analysis, they were identified as *B. taeanensis* SMI_1, *Enterobacter* sp. AABM_9, and *P. mendocina* AMPPS_5; both strains efficiently produced siderophores in Fe^+3^-deficient medium. To enhance siderophore production, the effect process parameters were studied. The highest siderophore production obtained was 93.57 %SU for SMI_1 after 48 h of incubation at 30 °C, pH 8, with sucrose as the carbon source, sodium nitrate as the nitrogen source, and 0.4% succinic acid. The optimum parameters for AABM_9 were 87.18 %SU after 36 h of incubation at 30 °C and pH 8 in the presence of sucrose, ammonium sulfate, and 0.4% succinic acid. Likewise, the maximum siderophore production achieved for AMPPS_5 was 91.17 %SU after 36 h of incubation at 35 °C, pH 8.5, with glucose, ammonium sulfate, and 0.4% citric acid as the organic acid. Both isolates showed growth and production of siderophores in the presence of metals, such as Fe^+3^, Cu^+2^, Mn^+2^, and Zn^+2^. However, upon increasing the concentration of Fe^+3^, siderophore production was radically decreased, possibly due to the negative regulation of transcriptional genes. Three isolates showed strong heavy-metal chelation activity in different heavy metals, and significant growth of peanut seedlings in seed germination experiments was observed in *Enterobacter* AABM_9. Further research is required to shed light on plant-growth-promoting marine bacteria and the mechanism involved in them.

## Figures and Tables

**Figure 1 microorganisms-11-02873-f001:**
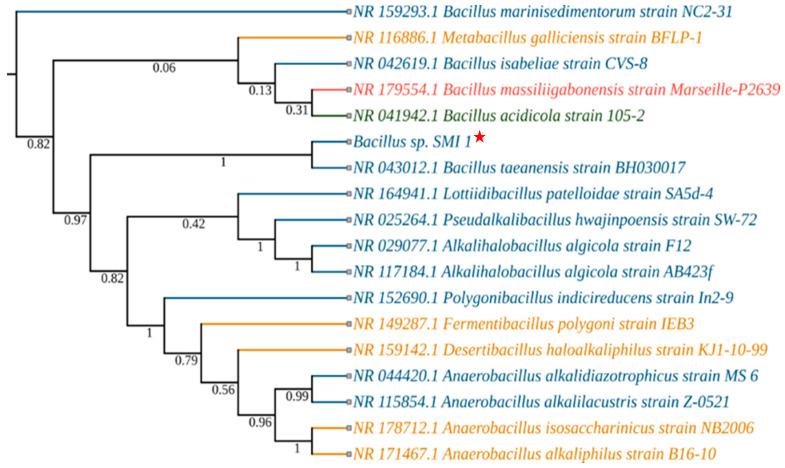
The phylogenetic tree of SMI_1 was constructed with the neighbor-joining method to analyze the evolutionary relationship of *Bacillus taeanensis* SMI_1. It showed 98% similarity with *Bacillus taeanensis* BH030017 based on NCBI (BLASTn) 16S rRNA sequences (the colors reflect the source of isolation: blue—marine; green—plant; red—human; yellow—soil and other). Red Star indicates the isolated marine bacterial isolate and its phylogenetic position.

**Figure 2 microorganisms-11-02873-f002:**
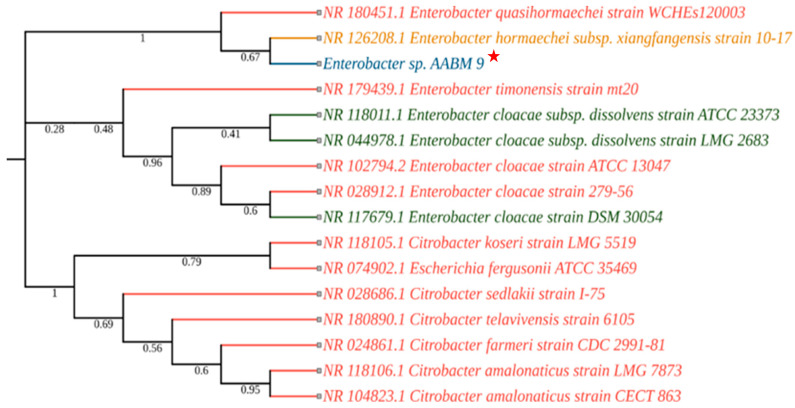
The neighbor-joining method was used to develop the phylogenetic tree to analyze the evolutionary relationship of *Enterobacter* sp. AABM_9, which showed 97.47% similarity with *Enterobacter hormaechei* based on NCBI (BLASTn) 16S rRNA sequences (the colors reflect the source of isolation: blue—marine; green—plant; red—human; yellow—soil and other). Red Star indicates the isolated marine bacterial isolate and its phylogenetic position.

**Figure 3 microorganisms-11-02873-f003:**
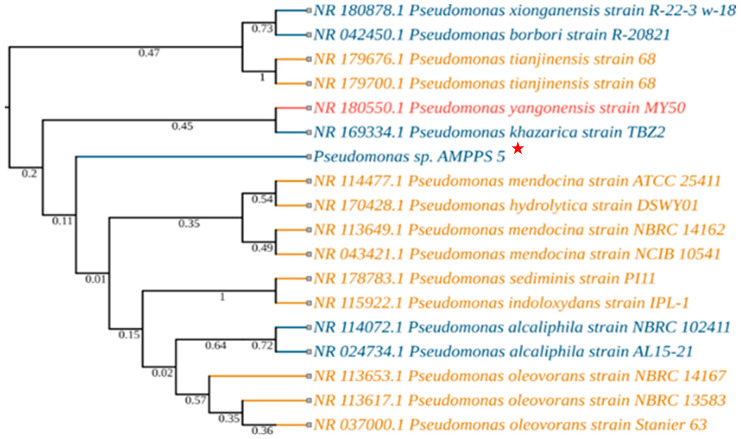
Evolutionary relationship and phylogenetic tree of AMPPS_5 with their closest strains of *Pseudomonas mendocina* based on NCBI (BLASTn) 16S rRNA sequences, showing 99% similarity. The neighbor-joining method was employed using MEGA-X software version 10.2.2 (the colors reflect the source of isolation: blue—marine; red—human; yellow—soil and other). Red Star indicates the isolated marine bacterial isolate and its phylogenetic position.

**Figure 4 microorganisms-11-02873-f004:**
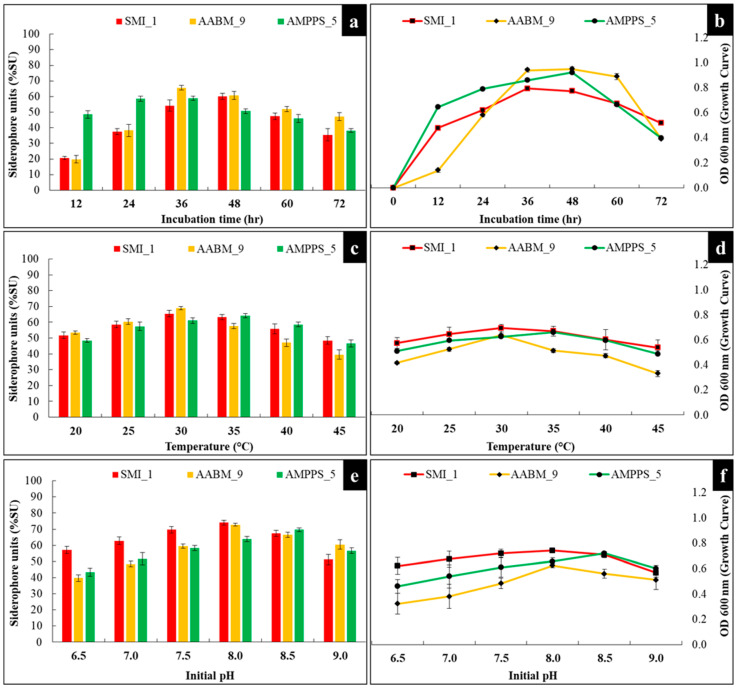
Effect of physicochemical parameters of (**a**,**b**) incubation time, (**c**,**d**) temperature, and (**e**,**f**) initial pH on siderophore production and growth of marine bacterial isolates (SMI_1, AABM_9, and AMPPS_5).

**Figure 5 microorganisms-11-02873-f005:**
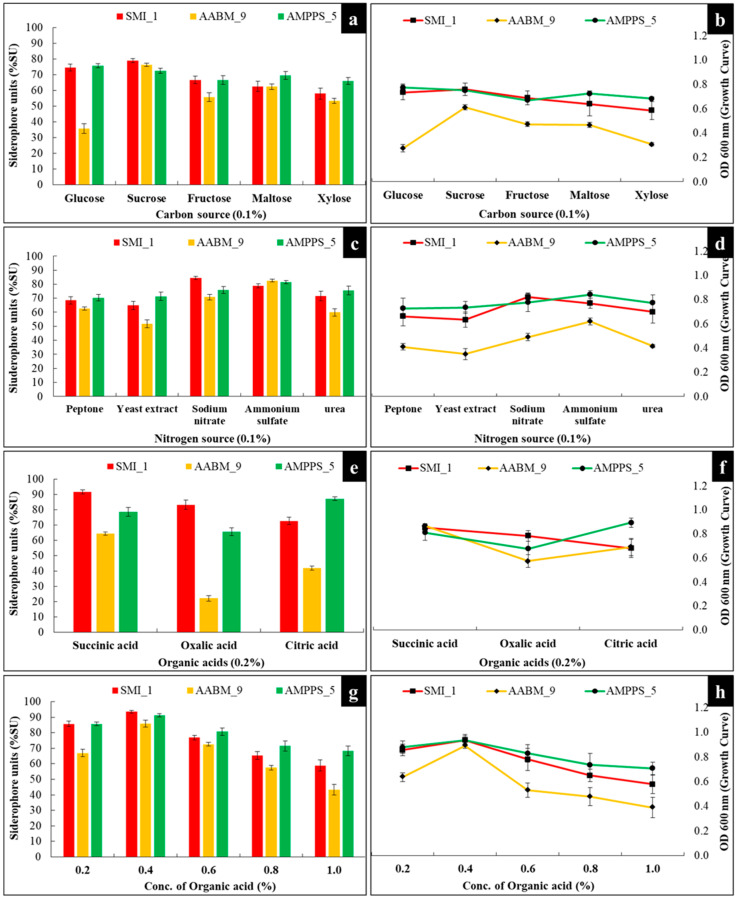
Effect of (**a**,**b**) carbon source, (**c**,**d**) nitrogen source, (**e**,**f**) organic acids, and (**g**,**h**) concentration of organic acid on siderophore production and growth of marine bacterial isolates (SMI_1, AABM_9, and AMPPS_5).

**Figure 6 microorganisms-11-02873-f006:**
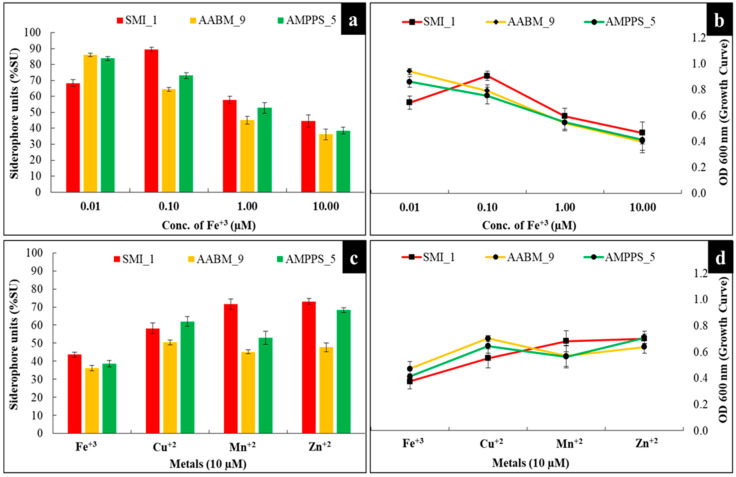
Effect of (**a**,**b**) different concentrations of Fe^+3^ and (**c**,**d**) different metal ions on siderophore production and growth of marine bacterial isolates (SMI_1, AABM_9, and AMPPS_5).

**Figure 7 microorganisms-11-02873-f007:**
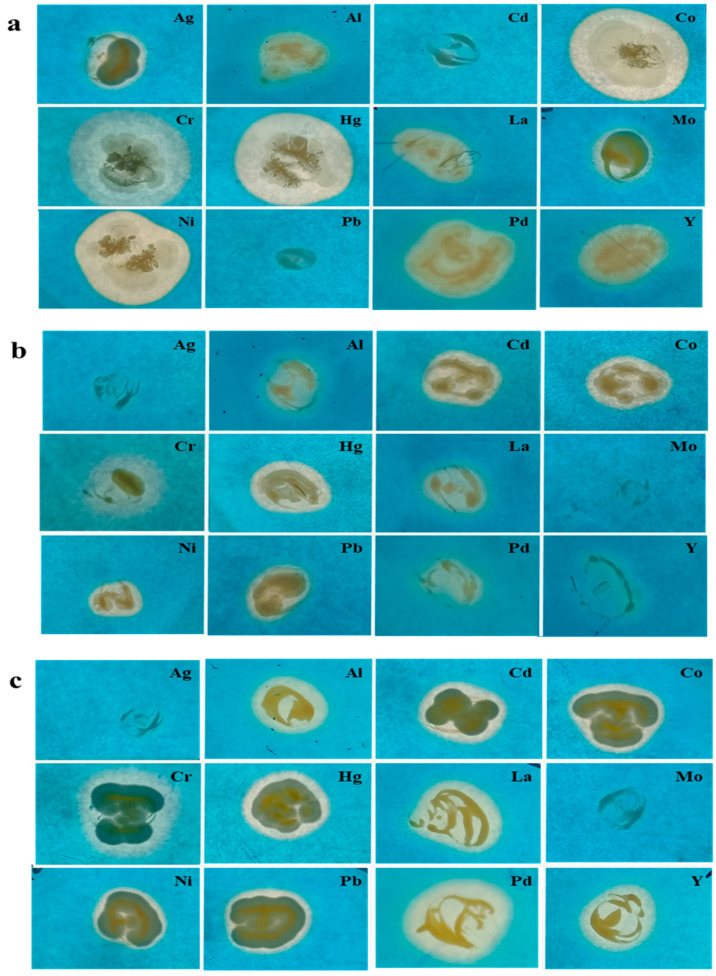
Heavy-metal ion chelation activity of marine bacterial isolates ((**a**) SMI_1, (**b**) AABM_9, and (**c**) AMPPS_5) observed through the CAS agar plate assay.

**Figure 8 microorganisms-11-02873-f008:**
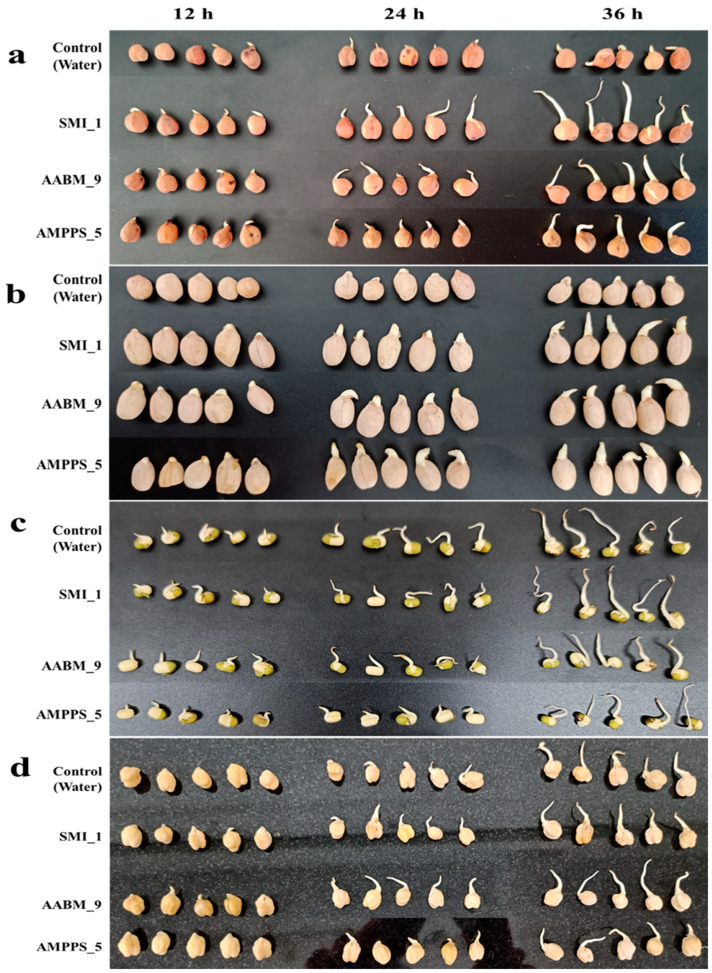
Seed germination ((**a**) Brown chickpea, (**b**) Peanut, (**c**) Green gram, and (**d**) Kabuli chana) in the presence of tap water (control) and cell-free supernatant of marine bacterial isolates (SMI_1, AABM_9, and AMPPS_5).

**Table 1 microorganisms-11-02873-t001:** Morphological, biochemical, and molecular characteristics of marine bacterial isolates (SMI_1, AABM_9, and AMPPS_5).

Characteristic(s)	Marine Bacterial Isolate(s)		
	SMI_1	AABM_9	AMPPS_5
Source	Marine sediment	Marine sediment	Marine water
Cell morphology	Gram-positive, rod-shaped	Gram-negative, rod-shaped	Gram-negative, rod-shaped
Colony morphology	Circular, cream, white, smooth edged, and slightly raised	Circular, smooth edged, convex, and light-yellow color	Smooth, flat, non-wrinkled, and pale brownish yellow
Hydrolysis of gelatin	Negative	Negative	Positive
Oxidase	Positive	Negative	Positive
Catalase	Positive	Positive	Positive
Closest relatives in NCBI GenBank	*Bacillus taeanensis*	*Enterobacter* sp.	*Pseudomonas mendocina*

**Table 2 microorganisms-11-02873-t002:** Effect of various physicochemical parameters on siderophore production (%SU) of marine bacterial isolate SMI_1.

Time	Temperature	Initial pH	Carbon Source	Nitrogen Source	Organic Acids	Conc. of Organic Acid	Conc. of Fe^+3^	Different Metals	SMI_1
(h)	(°C)		(0.1%)	(0.1%)	(0.2%)	(%)	(µM)	(10 µM)	(%SU)
12	28	7	-	-	Succinic acid	0.2	-	-	20.56 ± 1.09
24									37.62 ± 1.92
36									54.17 ± 3.61
48									60.17 ± 1.94
60									47.38 ± 2.05
72									35.54 ± 3.87
48	20	7	-	-	Succinic acid	0.2	-	-	51.73 ± 2.03
	25								58.52 ± 2.10
	30								65.45 ± 2.05
	35								63.19 ± 1.71
	40								55.91 ± 3.15
	45								48.51 ± 2.47
48	30	6.5	-	-	Succinic acid	0.2	-	-	57.13 ± 2.18
		7.0							62.83 ± 2.37
		7.5							69.66 ± 1.98
		8.0							74.04 ± 1.39
		8.5							67.42 ± 2.02
		9.0							51.20 ± 3.29
48	30	8.0	Glucose	-	Succinic acid	0.2	-	-	74.40 ± 2.23
			Sucrose						78.91 ± 1.38
			Fructose						66.68 ± 2.45
			Maltose						62.60 ± 3.15
			Xylose						57.97 ± 3.53
48	30	8.0	Sucrose	Peptone	Succinic acid	0.2	-	-	68.48 ± 2.66
				Yeast extract					64.87 ± 2.92
				Sodium nitrate					84.53 ± 1.17
				Ammonium sulfate					78.69 ± 1.45
				Urea					71.43 ± 3.43
48	30	8.0	Sucrose	Sodium nitrate	Succinic acid	0.2	-	-	91.74 ± 1.38
					Oxalic acid				83.35 ± 3.04
					Citric acid				72.64 ± 2.40
48	30	8.0	Sucrose	Sodium nitrate	Succinic acid	0.2		-	85.58 ± 1.79
						0.4			93.57 ± 0.91
						0.6			76.81 ± 1.57
						0.8			65.33 ± 2.45
						1.0			58.91 ± 3.43
48	30	8.0	Sucrose	Sodium nitrate	Succinic acid	0.4	0.01	-	68.35 ± 2.20
							0.10		89.45 ± 1.34
							1.00		57.85 ± 2.40
							10.0		44.62 ± 3.81
48	30	8.0	Sucrose	Sodium nitrate	Succinic acid	0.4	-	Fe^+3^	43.68 ± 1.36
								Cu^+2^	58.21 ± 2.94
								Mn^+2^	71.70 ± 2.83
								Zn^+2^	73.09 ± 1.68

**Table 3 microorganisms-11-02873-t003:** Effect of various physicochemical parameters on siderophore production (%SU) of marine bacterial isolate AABM_9.

Time	Temperature	Initial pH	Carbon Source	Nitrogen Source	Organic Acids	Conc. of Organic Acid	Conc. of Fe^+3^	Different Metals	AABM_9
(h)	(°C)		(0.1%)	(0.1%)	(0.2%)	(%)	(µM)	(10 µM)	(%SU)
12	28	7	-	-	Succinic acid	0.2	-	-	19.86 ± 2.32
24									38.31 ± 4.04
36									65.68 ± 1.43
48									60.81 ± 2.57
60									52.10 ± 1.68
72									47.23 ± 2.56
36	20	7	-	-	Succinic acid	0.2	-	-	53.38 ± 1.11
	25								60.36 ± 1.78
	30								68.88 ± 1.01
	35								57.59 ± 1.67
	40								47.08 ± 2.43
	45								39.57 ± 2.97
36	30	6.5	-	-	Succinic acid	0.2	-	-	39.76 ± 2.12
		7.0							48.48 ± 1.79
		7.5							59.45 ± 1.25
		8.0							72.76 ± 0.79
		8.5							66.57 ± 1.52
		9.0							60.50 ± 2.85
36	30	8.0	Glucose	-	Succinic acid	0.2	-	-	35.77 ± 3.16
			Sucrose						76.40 ± 1.06
			Fructose						55.68 ± 2.76
			Maltose						62.35 ± 1.67
			Xylose						53.22 ± 1.63
36	30	8.0	Sucrose	Peptone	Succinic acid	0.2	-	-	62.57 ± 1.05
				Yeast extract					51.69 ± 2.82
				Sodium nitrate					70.82 ± 1.98
				Ammonium sulfate					82.54 ± 1.10
				Urea					59.92 ± 2.62
36	30	8.0	Sucrose	Ammonium sulfate	Succinic acid	0.2	-	-	64.49 ± 1.01
					Oxalic acid				22.11 ± 1.83
					Citric acid				41.94 ± 1.47
36	30	8.0	Sucrose	Ammonium sulfate	Succinic acid	0.2	-	-	66.91 ± 2.40
						0.4			85.90 ± 2.32
						0.6			72.58 ± 1.27
						0.8			57.46 ± 1.53
						1.0			43.29 ± 3.36
36	30	8.0	Sucrose	Ammonium sulfate	Succinic acid	0.4	0.01	-	86.14 ± 1.08
							0.10		64.54 ± 1.23
							1.00		45.19 ± 2.47
							10.0		36.23 ± 3.45
36	30	8.0	Sucrose	Ammonium sulfate	Succinic acid	0.4	-	Fe^+3^	36.23 ± 1.45
								Cu^+2^	50.40 ± 1.33
								Mn^+2^	45.20 ± 1.05
								Zn^+2^	47.78 ± 2.43

**Table 4 microorganisms-11-02873-t004:** Effect of various physicochemical parameters on siderophore production (%SU) of marine bacterial isolate AMPPS_5.

Time	Temperature	Initial pH	Carbon Source	Nitrogen Source	Organic Acids	Conc. of Organic Acid	Conc. of Fe^+3^	Different Metals	AMPPS_5
(h)	(°C)		(0.1%)	(0.1%)	(0.2%)	(%)	(µM)	(10 µM)	(%SU)
12	28	7	-	-	Succinic acid	0.2	-	-	48.54 ± 2.52
24									58.58 ± 1.57
36									58.86 ± 1.30
48									50.68 ± 1.43
60									45.94 ± 2.64
72									38.27 ± 1.25
36	20	7	-	-	Succinic acid	0.2	-	-	48.50 ± 1.28
	25								57.37 ± 2.68
	30								61.08 ± 1.72
	35								64.05 ± 1.30
	40								58.64 ± 1.56
	45								46.60 ± 2.34
36	35	6.5	-	-	Succinic acid	0.2	-	-	43.39 ± 2.44
		7.0							51.66 ± 3.83
		7.5							58.36 ± 1.54
		8.0							63.90 ± 1.61
		8.5							69.65 ± 1.10
		9.0							56.66 ± 1.68
36	35	8.5	Glucose	-	Succinic acid	0.2	-	-	75.69 ± 1.28
			Sucrose						72.38 ± 1.56
			Fructose						66.44 ± 2.78
			Maltose						69.58 ± 2.51
			Xylose						66.05 ± 2.16
36	35	8.5	Glucose	Peptone	Succinic acid	0.2	-	-	70.35 ± 2.19
				Yeast extract					71.32 ± 2.90
				Sodium nitrate					75.71 ± 2.44
				Ammonium sulfate					81.40 ± 1.26
				Urea					75.47 ± 3.12
36	35	8.5	Glucose	Ammonium sulfate	Succinic acid	0.2	-	-	78.59 ± 2.87
					Oxalic acid				65.66 ± 2.61
					Citric acid				87.25 ± 1.23
36	35	8.5	Glucose	Ammonium sulfate	Citric acid	0.2	-	-	85.65 ± 1.15
						0.4			91.17 ± 0.96
						0.6			80.70 ± 2.48
						0.8			71.43 ± 3.33
						1.0			68.26 ± 3.04
36	35	8.5	Glucose	Ammonium sulfate	Citric acid	0.4	0.01	-	83.77 ± 1.19
							0.10		73.10 ± 1.87
							1.00		52.80 ± 3.29
							10.0		38.61 ± 2.06
36	35	8.5	Glucose	Ammonium sulfate	Citric acid	0.4	-	Fe^+3^	38.57 ± 1.72
								Cu^+2^	62.01 ± 2.82
								Mn^+2^	52.97 ± 3.70
								Zn^+2^	68.26 ± 1.35

**Table 5 microorganisms-11-02873-t005:** Optimized production parameters and siderophore production were compared with the published literature.

Micro-Organism	Habitat	Incubation Time (h)	Temp (°C)	pH	Carbon Source	Nitrogen Source	Organic Acid	Yield	Ref.
*P. aeruginosa* FP6	Soil sample	-	-	-	Sucrose Mannitol	Urea	-	104.8 mM 92.9 mM	[43]
*B. cereus* *P. weihenstephanensis*	Marine	100–150	25	8.5	-	-	-	-	[44]
*Brevibacillus brevis* GZDF3	Rhizosphere soil	48	32	7	Sucrose	Asparagine	-	-	[6]
*P. aeruginosa* RZS9	-	24	27.8	7.1	-	-	Succinic acid	69.03 %SU	[45]
*Bacillus* sp. PZ-1	Soil sample	48	30	6.2	Glucose	Asparagine		90.52 %SU	[46]
*P. fluorescens*	-	24	29	7	Glucose	Urea	Succinic acid	96%	[47]
*Bacillus* sp. (VITVK5) *Enterobacter* sp. (VITVK6)	Soil sample	-	37	8	Sucrose Glucose	Sodium Nitrate	Citric acid	~60–80%	[48]
*E. coli* *Bacillus* spp. ST13 *Streptomyces pilosus*	-	*-*	55	6	Sucrose Glucose	-	-	48 μg/mL 31 μg/mL 32 μg/mL	[49]
*P. aeruginosa*	-	-	27.8	7.1	-	-	-	68.41%	[50]
*P. aeruginosa* azar 11	Aquatic soil	72	37	7	Maltose	Ammonium nitrate	Citric acid	59.18%	[51]
*Marinobacter hydrocarbonoclausticus* SVU_3	Marine	48	30	8.5	Glucose	Sodium nitrate	Succinic acid	82.75 %SU	[52]
*B. teanensis* SMI_1	Marine	48	30	8	Sucrose	Sodium nitrate	Succinic acid	96.48 %SU	Present study
*Enterobacter* sp. AABM_9	Marine	36	30	8	Sucrose	Ammonium sulfate	Succinic acid	86.14 %SU	Present study
*P. mendocina* AMPPS_5	Marine	36	35	8.5	Glucose	Ammonium sulfate	Citric acid	91.17 %SU	Present study

**Table 6 microorganisms-11-02873-t006:** Heavy-metal chelation activity of marine bacterial isolates (SMI_1, AABM_9, and AMPPS_5) observed through the CAS agar plate assay (- indicates no growth; + indicates growth; ++ indicates growth with low activity; +++ indicates growth with moderate-to-good activity).

Metal Salt(s) (1 mM)	Metal Ion(s)	Marine Bacterial Isolate(s)		
		SMI_1	AABM_9	AMPPS_5
FeCl_3_.7H_2_O	Fe^+3^	+++	++	++
AgNO_3_	Ag^+2^	++	-	-
Al_2_(SO_4_)_3_	Al^+3^	++	+++	++
CdCl_2_	Cd^+2^	-	++	++
CoCl_2_.6H_2_O	Co^+2^	+++	++	++
K_2_Cr_2_O_7_	Cr^+6^	+++	++	++
HgCl_2_	Hg^+2^	+++	++	++
La_2_O_3_	La^+3^	++	+++	++
Na_2_MoO_4_.2H_2_O	Mo^+6^	++	-	-
NiCl_2_.6H_2_O	Ni^+2^	+	-	-
C_4_H_6_O_4_Pb.3H_2_O	Pb^+2^	-	++	++
PdCl_2_	Pd^+2^	++	+++	+
Y_2_O_3_	Y^+3^	++	+++	+

**Table 7 microorganisms-11-02873-t007:** Length of the seed (Brown chickpea, Peanut, Green gram, and Kabuli chana) germinated in the presence of tap water (control) and cell-free supernatant of marine bacterial isolates (SMI_1, AABM_9, and AMPPS_5) at various incubation times (12 h, 24 h, and 36 h).

Species	Cell-Free Supernatant	Length (cm) at Incubation Time (h)			% GP
		12	24	36	
Brown chickpea (*Cicer arietinum* L.)	Control (Tap Water)	0	0.16 ± 0.16	0.38 ± 0.08	36.2
	SMI_1	0.32 ± 0.13	0.82 ± 0.22	1.76 ± 0.35	92.4
	AABM_9	0.23 ± 0.04	0.6 ± 0.12	1.28 ± 0.14	89.1
	AMPPS_5	0.14 ± 0.08	0.36 ± 0.05	0.84 ± 0.13	67.3
Peanut (*Arachis hypogaea*)	Control (Tap Water)	0	0.18 ± 0.11	0.42 ± 0.08	28.1
	SMI_1	0.22 ± 0.04	0.52 ± 0.08	1.12 ± 0.13	84.9
	AABM_9	0.24 ± 0.05	0.46 ± 0.09	0.74 ± 0.20	71.4
	AMPPS_5	0.14 ± 0.05	0.66 ± 0.16	0.9 ± 0.12	63.2
Green gram (*Vigna radiata*)	Control (Tap Water)	0.26 ± 0.08	0.94 ± 0.36	1.8 ± 0.33	67.4
	SMI_1	0.34 ± 0.13	0.74 ± 0.11	2.22 ± 0.4	96.2
	AABM_9	0.32 ± 0.08	0.86 ± 0.86	1.58 ± 0.19	93.7
	AMPPS_5	0.28 ± 0.08	0.58 ± 0.08	1.54 ± 0.33	81.3
Kabuli chana (*Cicer arietinum*)	Control (Tap Water)	0	0.98 ± 0.39	1.96 ± 0.4	52.6
	SMI_1	0.28 ± 0.13	0.92 ± 0.27	1.86 ± 0.39	89.2
	AABM_9	0.22 ± 0.08	1.4 ± 0.29	2.04 ± 0.37	82.5
	AMPPS_5	0.04 ± 0.05	0.6 ± 0.2	1.44 ± 0.4	78.3

## Data Availability

All the data are presented in the paper.

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
