# Peer review of "Optimization of Siderophore Production in Three Marine Bacterial Isolates along with Their Heavy-Metal Chelation and Seed Germination Potential Determination"

_microorganisms, 2023, doi:10.3390/microorganisms11122873_

Round 1
Reviewer 1 Report
Comments and Suggestions for Authors
Dear Authors
the idea is great, the work extensive, but the focus is weak; I suggest to put the focus on application of the bacterial strains for seed support. Then many details on the siderophore production may move to supplemental material. This will make it clear and readable.
Also there is some work in similar direction for soil Actinobacteria and reviews summarizing this which are not mentioned; so I suggest you check the literature again.
minor points
- Line 70 to 80; the redox state of Fe need to be checked.
- methods; correct media to medium were appropriate.
- the legends of the phylogenetic trees in the result section need more details.
- section 3.2 has no results, it is more a state of the art?
- Siderophore production is given only in %; not outline what it is, how it related to a concentration or reference; this needs to be outline and mentioned at tables as well.
- was the glass ware washed prior use to remove all traces of iron?
Comments on the Quality of English Language
Language is readable and can be corrected for minor mistakes.
Author Response
Authors thank the reviewer for their invaluable and constructive suggestions.

Reviewer 2 Report
Comments and Suggestions for Authors
This study optimized siderophores production in three marine bacterial isolates along with their heavy metal chelation. The optimization procedures shown in Tables 1-3 are excellent and convincing. Also, seed germination potential determination of siderophores in this study was interesting.
But, describe what mechanisms was supposed for siderophores in the germination tests in Introduction section. Seed germination ratio was not affected by siderophores (L379-381). According to Table 7, siderophores did not enhance the plumule length of green gram and kabul chana. This was contradictory to description on L382-383 and conclusion on L611-613. On L 383, it was described that “germination percentage of each seed in respective isolate supernatant were reported in Table 7”. But no data of germination percentage on Table 7. Regrettably, experimental results were not understood correctly. Discuss why siderophores was effective for only brown chick pea and peanut.
Also, there have been many researches on siderophores production in bacterial species. The originality of this study is unclear. On L80-82, the authors described that “compared to terrestrial siderophores, most of the marine siderophores show unique structural properties. In view of that, they show photochemical and amphipathic properties in Fe+3 complexes”. This should be described more clearly and discussed in the Discussion part. Discuss what is advantages of siderophores of the bacterial isolates in this study.
The discussion section only introduced or reviewed other studies. These can be written in the introduction section rather than in the discussion section. The authors should claim the novelty and usefulness of the results obtained in this study compared to those studies.
Discussion, the first paragraph (L395-399) should be partly moved to the results section (ZMA media was used to isolate sixty-eight marine bacteria from Kalinga beach, Bay of Bengal. Among them, 70% of the isolates had siderophore producing ability, which was confirmed by observing “orange halo zones” on blue agar plates.).
Discussion, the third paragraph (L408-420) was just repeat of methodology and results.
Discussion, the forth paragraph (L421-432) was just introduction of methodology for siderophore studies enough to be understood without results of this study.
Related to the fifth and sixth paragraph (L442-481) of Discussion, what was the temperature and pH range of the origin of bacteria in this study?
Discussion, the 13th paragraph (L549-561), this was just introduction of metal pollution risks enough to be understood without results of this study.
Discussion, the 14-15th paragraph (L572-591), this was just introduction of plant growth by bacteria enough to be understood without results of this study.
“6. Economical Aspects” and “7. Future Research Prospects” were as not necessary because these contents were general for siderophore studies enough to be understood without this study.
Author Response
Authors thank the reviewer for their constructive comments.

Round 2
Reviewer 2 Report
Comments and Suggestions for Authors
QA2 .> As stated, the suggested details were mentioned in the paragraph of the discussion.
Some marine siderophoreres may be amphiphilic with a series of fatty acid attachments, have α-hydroxycarboxylic acid head group with Fe3+, triscatechol with catechol DHB. Are these characteristics common to the siderophores studied in this research?
QA3. As suggested, few sections have been moved to introduction section, and the novelty and usefulness of the study was stated in the last paragraph of the discussion.
The last paragraph of the discussion which was added in the revised manuscript are not necessary. Its content should be included in the conclusion. This should be the first report on siderophore production from your strains SMI_1, AABM_9 and AMPPS_5. But there's no point in describing that. Have there been any reports on siderophore production from Bacillus, Enterobacter sp., Pseudomonas?
QA7. As stated, suggested information was included in the fifth and sixth paragraphs of discussion. The isolated bacteria were primarily incubated at 30 °C.
My question was “what was the temperature and pH range of the origin of bacteria in this study?”. The origin of those bacteria were Kalinga beach, Bengal Bay, Visakhapatnam?
Author Response
Response to Reviewer Comments QA2. As stated, the suggested details were mentioned in the paragraph of the discussion. Some marine siderophores may be amphiphilic with a series of fatty acid attachments, have α-hydroxycarboxylic acid head group with Fe3+, triscatechol with catechol DHB. Are these characteristics common to the siderophores studied in this research? As stated, specific characteristics or nature of the siderophore should be determined. However, the work reported in the present manuscript is preliminary chemical characterization tests (L303-307). Further work, i.e., to identify exact structure and chemical nature of the marine siderophores, high-throughput techniques like LC-MS/MS and NMR will be employed. At present, method development for purification through HPLC is going on, which will be followed by biochemical and biophysical characterization studies of the purified siderophore. QA3. As suggested, few sections have been moved to introduction section, and the novelty and usefulness of the study was stated in the last paragraph of the discussion. The last paragraph of the discussion which was added in the revised manuscript are not necessary. Its content should be included in the conclusion. This should be the first report on siderophore production from your strains SMI_1, AABM_9 and AMPPS_5. But there's no point in describing that. Have there been any reports on siderophore production from Bacillus, Enterobacter sp., Pseudomonas? As suggested, the paragraph has been removed, and studies on siderophore production from Bacillus, Enterobacter sp., and Pseudomonas have been added. Please go through the following references. 1. Wilson, M.K.; Abergel, R.J.; Raymond, K.N.; Arceneaux, J.E.L.; Byers, B.R. Siderophores of Bacillus Anthracis, Bacillus Cereus, and Bacillus Thuringiensis. Biochem. Biophys. Res. Commun. 2006, 348, 320–325, doi:10.1016/j.bbrc.2006.07.055. 2. Chakraborty, K.; Kizhakkekalam, V.K.; Joy, M.; Chakraborty, R.D. Bacillibactin Class of Siderophore Antibiotics from a Marine Symbiotic Bacillus as Promising Antibacterial Agents. Appl. Microbiol. Biotechnol. 2022, 106, 329–340, doi:10.1007/s00253-021-11632-0. 3. Sinha, A.K.; Parli, B. V. Siderophore Production by Bacteria Isolated from Mangrove Sediments: A Microcosm Study. J. Exp. Mar. Bio. Ecol. 2020, 524, 151290, doi:10.1016/j.jembe.2019.151290. 4. Murugappan, R.M.; Aravinth, A.; Rajaroobia, R.; Karthikeyan, M.; Alamelu, M.R. Optimization of MM9 Medium Constituents for Enhancement of Siderophoregenesis in Marine Pseudomonas Putida Using Response Surface Methodology. Indian J. Microbiol. 2012, 52, 433–441, doi:10.1007/s12088-012-0258-y. QA7. As stated, suggested information was included in the fifth and sixth paragraphs of discussion. The isolated bacteria were primarily incubated at 30 °C. My question was “what was the temperature and pH range of the origin of bacteria in this study?”. The origin of those bacteria was Kalinga beach, Bengal Bay, Visakhapatnam? As stated, the temperature and pH of origin of bacteria was included in L245.
